# Arbitrary aperture synthesis with nonlocal leaky-wave metasurface antennas

Gengyu Xu[1], Adam Overvig [1], Yoshiaki Kasahara[1,2], Enrica Martini [3], Stefano Maci[3] & Andrea Alù [1,4,5] ✉

The emergence of new technological needs in 5 G/6 G networking and broadband satellite internet access amplifies the demand for innovative wireless communication hardware, including high-performance low-profile transceivers. In this context, antennas based on metasurfaces – artificial surfaces engineered to manipulate electromagnetic waves at will – represent highly promising solutions. In this article, we introduce leaky-wave metasurface antennas operating at micro/millimeter-wave frequencies that are designed using the principles of quasi-bound states in the continuum, exploiting judiciously tailored spatial symmetries that enable fully customized radiation. Specifically, we unveil additional degrees of control over leaky-wave radiation by demonstrating pointwise control of the amplitude, phase and polarization state of the metasurface aperture fields by carefully breaking relevant symmetries with tailored perturbations. We design and experimentally demonstrate metasurface antenna prototypes showcasing a variety of functionalities advancing capabilities in wireless communications, including single-input multi-output and multi-input multi-output near-field focusing, as well as far-field beam shaping.

Leaky wave antennas (LWAs) are traveling wave transceivers with highly desirable practical features, including high directivity, low profile, simple feeding architecture, and inherent beam-scanning capability enabled by frequency dispersion[1]. They leverage tailored perturbations that partially couple guided electromagnetic modes into directive free-space radiation. Owing to their flexibility, LWAs have found applications not just in wireless communications but also for imaging[2,3], radar detection[4], and remote sensing[5] systems. The key design parameters for a traditional LWA are the attenuation and propagation constants, which respectively control, to a limited degree, the effective aperture and scan angle of the radiated beam (typically a simple linearly-polarized plane wave). The polarization is usually fixed to a single state, dictated by the antenna architecture.

The limited set of degrees of freedom (DoFs) offered by conventional LWAs does not meet the great demand from today's wireless infrastructures for data capacity, channel diversity, and energy efficiency. Hence, recent years have witnessed a proliferation of advanced antenna designs based on electromagnetic metamaterials and metasurfaces[6–10]. These artificial structures consist of subwavelength building blocks called "meta-atoms" or "meta-units" with engineered electromagnetic scattering responses. Their ability to manipulate fields and waves has unlocked a plethora of high-performance radio-frequency wireless devices, including LWAs, with unprecedented capabilities. For example, ultra-thin artificial impedance surfaces supporting surface waves can be made radiating via periodic or quasi-periodic perturbations to their effective electric impedance, forming holographic leaky-wave metasurfaces (LWMs)[11,12]. With properly engineered anisotropy, the effective impedance takes a tensorial form, leading to radiation with controlled polarization states[13,14]. To exert simultaneous and independent control over all aspects of the aperture

[1]Photonics Initiative, Advanced Science Research Center, City University of New York, New York, NY 10031, USA. [2]Department of Electrical Engineering, The University of Texas at Austin, Austin, TX 78712, USA. [3]Department of Information Engineering and Mathematics, University of Siena, Siena 53100, Italy. [4]Department of Electrical Engineering, The City College of New York, New York, NY 10031, USA. [5]Physics Program, Graduate Center, City University of New York, New York, NY 10016, USA. ✉e-mail: aalu@gc.cuny.edu

field and, accordingly, the far-field radiation, LWAs incorporating Huygens' metasurfaces[15–17] or cascaded tensorial impedance sheets have been proposed[18]. Instead of modeling the meta-atoms as continuous sheets of effective electric and magnetic polarization currents, it is also possible to treat them as discrete arrays of non-interacting resonant dipoles[19–21]. This latter approach trades some of the beam-forming DoFs of the former for simpler designs that are easier to realize in practice.

Besides the aforementioned metasurfaces exploiting an engineered local response to external fields, the emerging field of "nonlocal" metasurfaces has been showing that it is possible to fully tailor all aspects of electromagnetic wave interactions by harnessing the mutual coupling among meta-units (e.g., mediated by a guided mode). These nonlocal metasurfaces have been receiving intense investigation in the optical regime, particularly in the context of bound states in the continuum (BICs), which are resonant modes that remain confined despite their compatibility in terms of momentum with a continuum of radiating waves (i.e., accessible radiation channels)[22,23]. One class of BICs, known as symmetry-protected BICs, is of particular interest to the scientific and engineering communities[23,24]. They arise from certain symmetries in the modal field profile that inhibit coupling to far-field radiation due to mismatched field parity. However, when these modal symmetries are broken through tailored perturbations, they turn into "quasi-BICs" (q-BICs), which can radiate with fully controlled features[25,26]. Researchers have utilized this concept to shape the wavefront, polarization, and spectra of coherent as well as thermal emission[27] with extreme precision. Very recently, this concept has also enabled integrated photonic LWMs leveraging spatially varying perturbations to symmetry-broken photonic crystal slabs to convert guided optical modes into customized free-space emission[28].

The q-BIC framework is very powerful in terms of added DoFs and waveform control since its underlying principles stem from spatial symmetries[25]. Hence it is generalizable to any frequency regime, and it is agnostic to details of the physical implementation (to first order in perturbation theory). Furthermore, it enables a completely rational design scheme for highly sophisticated nonlocal metasurfaces by providing a set of simple algebraic relations linking the four DoFs of a monochromatic wave (amplitude, phase, orientation, and ellipticity of polarization) to four independent geometrical DoFs embedded inside the constituent meta-unit. This feature represents a significant advantage over conventional metasurface design approaches, which often require cumbersome numerical optimization or the construction of extensive meta-atom look-up tables to bridge the gap between idealized analytical models and physical implementations.

In this work, motivated by emerging technological needs in 5 G/6 G networking, satellite internet, autonomous vehicles, and smart/cognitive radio environments[29,30], we translate the concept of q-BICs from optical frequencies into the micro/millimeter-wave spectrum to empower advanced nonlocal LWM antennas with fully controlled radiation features. Although the underlying design principles are consistent with integrated photonic LWMs[28], operation at mm-waves enables increased flexibility by the much more versatile and accessible printed circuit board (PCB) manufacturing platform. For example, the incorporation of a conductive ground plane admits unidirectional radiation with significantly higher directivity than attainable at optical frequencies (for which compact and low-loss reflectors near the guided mode are not trivially achieved). Furthermore, we can feed the LWM with sophisticated and compact substrate-integrated wave launchers, opening the opportunity for more advanced modes of operation (e.g., multi-port excitations). Moving into radio frequencies also introduces new challenges, such as limitations on the aperture size and the overall device footprint (due to increased wavelength), demanding innovative solutions that take full advantage of the available fabrication technologies.

As proofs of concept, we present three q-BIC-based LWM antenna prototypes with unprecedented functionalities for leaky-wave

radiation control. The first metasurface antenna is a single-input multiple-output (SIMO) LWM lens that generates two distinct orthogonally polarized focal spots placed at arbitrary locations from a single guided mode input. The second design realizes a multiple-input multiple-output (MIMO) lens that focuses its radiation to different locations within the Fresnel zone, depending on which of its two ports is excited. Lastly, we present a multi-beam LWM antenna with dual-polarized far-field beam-shaping capabilities. All three implementations are validated through full-wave numerical simulations, as well as experimental measurements.

## Results
### Working principle

The physical principles governing the radiative behavior of symmetry-protected q-BICs are general, and they can be applied to a range of vastly different device platforms. Here, for illustrative purposes, we utilize a parallel plate waveguide (PPWG) with thickness $h$ and relative permittivity $\epsilon_r$, perforated by a staggered array of rectangular slots (Fig. 1), which supports a q-BIC in the form of a quasi-transverse-electromagnetic (TEM) mode. The shape and the orientation of the slots have been tailored such that the mode leaks energy in a fully controlled fashion as it propagates. Here, the parameters $A$ and $\Phi$ refer to the local field amplitude and phase over the LWM aperture, while $2\psi$ and $2\chi$ refer to the longitude and latitude of the local polarization state on the Poincaré sphere. The dependence of these parameters on the spatial coordinates $(x,y)$ and a global time convention of $e^{j\omega t}$ are left implicit for brevity. In the following subsections, we summarize the design methods employed to rationally pattern the slot array and exert pointwise control over all four DoFs of the aperture field, along with physical insights into the underlying phenomena.

### Control of radiation magnitude

We begin by examining the simplest case of an LWM perforated by uniformly sized staggered square slots (with width $w$), whose primitive unit-cell (when considering a rectangular lattice) is pictured in Fig. 2a, marked by the dotted outline and with dimensions $L_x \times 2L_y$. Using numerical simulations (see "Methods"), we evaluated the first Brillouin Zone (FBZ) of a typical dispersion diagram for $x$-directed wave vectors

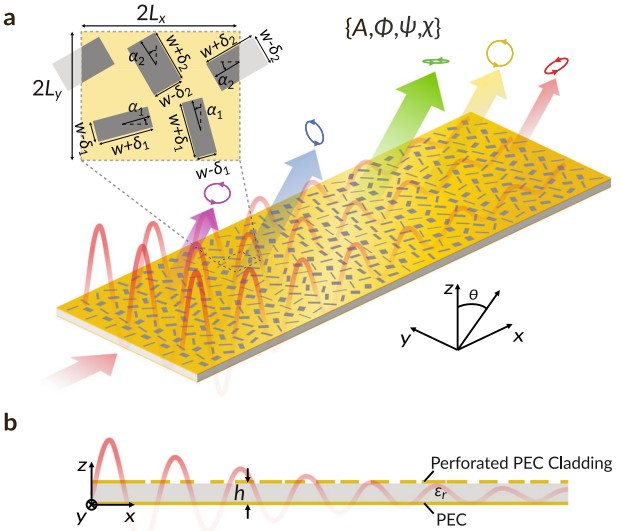

**Fig. 1 | Leaky-wave metasurface antenna supporting quasi-bound states in the continuum. a** Isometric and **b** side view of a schematic drawing. The amplitude $A$, phase $\Phi$ and polarization state $\psi$, $\chi$ of the aperture fields can be independently controlled in a point-wise fashion by locally engineering the geometrical perturbations (inset) to the perforated perfect electric conductor (PEC) parallel plate waveguide.

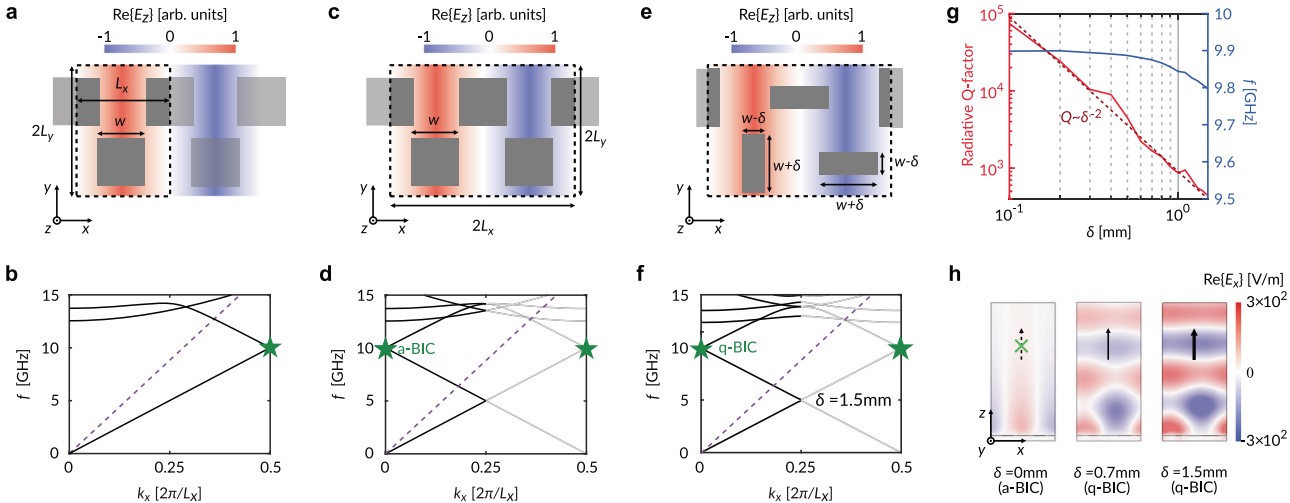

**Fig. 2 | Quasi-bound state in the continuum with controlled radiative Q-factor.**
**a** Sketch of the electric field of a bound quasi-TEM mode supported by the perforated waveguide and **b** its simulated dispersion diagram, assuming $2L_x = 8.5\,\text{mm}$, $L_y = 6.5\,\text{mm}$, $w = 3\,\text{mm}$, $\epsilon_r = 3$, $h = 1.52\,\text{mm}$. The purple dashed line corresponds to the free-space light line. **c** Artificial periodic doubling along the $\hat{x}$ direction corresponds to **d** folding the FBZ, which moves the mode of interest (green star) to the Γ

point in an artificial continuum. **e** A period-doubling, symmetry-breaking perturbation with strength $\delta$ converts the mode from an artificial BIC to a quasi-BIC with minimal disturbance to the **f** band structure. **g** The radiative Q-factor of the q-BIC varies as $\delta^{-2}$, while its resonant frequency remains mostly constant for small values $\delta$. **h** Simulated electric field intensity above the waveguide, showing stronger radiation due to larger perturbation.

(Fig. 2b), in which the purple dashed line represents the free-space light line. The lowest-order eigenmode is a quasi-TEM traveling wave with respect to $\hat{x}$. At the X high-symmetry point ($k_x = \pi/L_x$) of the FBZ, such a mode (marked by the green star) can be referred to as an "artificial BIC" (a-BIC)[31]: when we artificially double the metasurface periodicity along $\hat{x}$ by considering two adjacent primitive cells as a collective meta-unit, as done in Fig. 2c, the mode is shifted to the Γ point ($k_x = 0$) of the folded FBZ (Fig. 2d), where it now resides above the light line. Despite artificially coexisting with a continuum of radiating waves, this mode cannot couple to the far field. One interpretation for the lack of energy leakage is provided by the symmetry of the modal field profile: since the aperture electric field over every two adjacent slots have the same magnitude but are in anti-phase, they destructively interfere with each other in the far field. As a result, the a-BIC possesses an ideally infinite radiative Q-factor. Ultimately, it is important to remember that, in the current configuration, the mode, in fact, resides below the light line. Therefore, its confinement to the waveguide is not surprising, and it is guaranteed by translational symmetry.

The key realization to enable LWMs is that we can revoke the symmetry protection that inhibits the a-BIC from radiating by introducing a period-doubling symmetry-breaking perturbation to the meta-unit as depicted in Fig. 2e (and with the corresponding band diagram in Fig. 2f), where the parameter $\delta$ quantifies the degree of introduced asymmetry. In this way, one of the two neighboring slots radiates stronger than the other, and the mode is transformed into a leaky q-BIC with a finite radiative Q-factor. Hence, the "artificial continuum" is promoted to a "true continuum". As seen in Fig. 2g, the Q-factor varies approximately as $\delta^{-2}$, implying that a higher degree of asymmetry leads to more intense radiation[23,24]. To illustrate this further, Fig. 2h showcases snapshots of the simulated electric field distribution above the waveguide for several values of $\delta$. Larger perturbations evidently produce stronger radiated fields. It is also possible to map the Q-factor to the corresponding attenuation (leakage) constant, more commonly used in the leaky-wave antenna literature[32].

The degree of asymmetry of each meta-unit can be individually controlled, allowing us to precisely pattern the amplitude profile of the entire LWM aperture in a point-wise fashion. In this study, we have neglected the tapering of the q-BIC power density as it propagates along the LWM. Such an approximation will not cause significant issues for weakly perturbed designs since they generally have low leakage

rates. It is also worth noting that, as revealed by Fig. 2g, the resonant frequency of the q-BIC, i.e., the eigenfrequency at the Γ point, is only weakly dependent on the perturbation strength. When larger perturbations are considered, it may be necessary to slightly adjust the slot dimensions in order to compensate for the frequency drift, which may otherwise lead to undesirable radiation squinting.

## Control of linear polarization state
The polarization of the radiated field can be customized in a pointwise manner by breaking the mirror symmetries of each adjacent pair of slots. For instance, we can introduce a rotation angle $\alpha$, as shown in the inset of Fig. 3. When $\alpha \in [0°, 90°]$, the aperture electric fields of both slots are symmetric across the $xz$-plane. A magnetic wall bisects both slots, leading to purely $x$-polarized radiation into the far field. Conversely, when $\alpha \in [45°, 135°]$, a magnetic wall is formed between the slot pair, giving rise to purely $y$-polarized radiation. Continuously sweeping $\alpha$ from 0° to 180° allows us to traverse the equator of the Poincaré sphere twice, covering all possible combinations of linear polarization states and polarity.

## Control of aperture phase
To locally control the phase of the radiated fields, we introduce another geometric degree of freedom, imparting different perturbation strengths to the lower and upper rows of the meta-unit. At the Γ point, the modal profile of the q-BIC includes two rows of radiating apertures that are 90° out of phase, as they are staggered by 1/4 of a guided wavelength. Hence, the perturbation strengths for the lower and upper pairs ($\delta_1$ and $\delta_2$) can independently control the in-phase and quadrature components of the radiating part of the aperture field, granting us full 360° phase coverage. In Fig. 4, we demonstrate how to tailor the phase for $x$-polarized fields ($\Phi_x$). The same scheme can be used for any other linearly polarized radiation if we set $\alpha \notin [0°, 90°]$, following Fig. 3.

We can parameterize the perturbation strengths of the two rows as

$$\delta_1 = \delta\cos\Phi, \quad \delta_2 = \delta\sin\Phi \tag{1}$$

where $\Phi$ is the desired phase. This mapping reveals the geometric nature of the aperture phase, implying that, when patterned with slowly varying profiles, the LWM is robust against performance

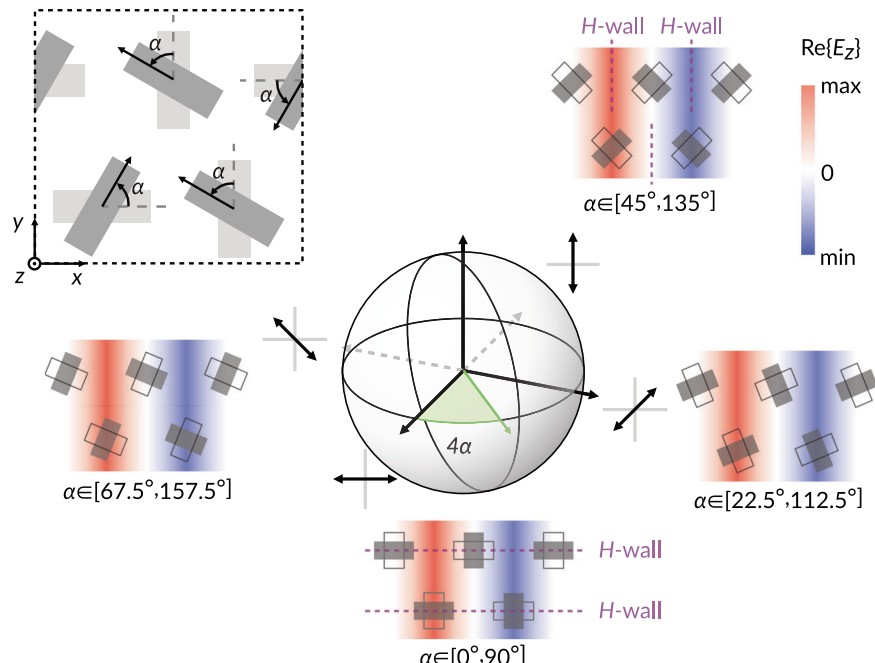

**Fig. 3 | Control of the linear polarization state of radiation through rotation of the slots.** The two choices of $\alpha$ (differing by $\pi/2$) are depicted as gray solid rectangles and empty rectangles; the fields produced differ by a phase factor $\pi$.

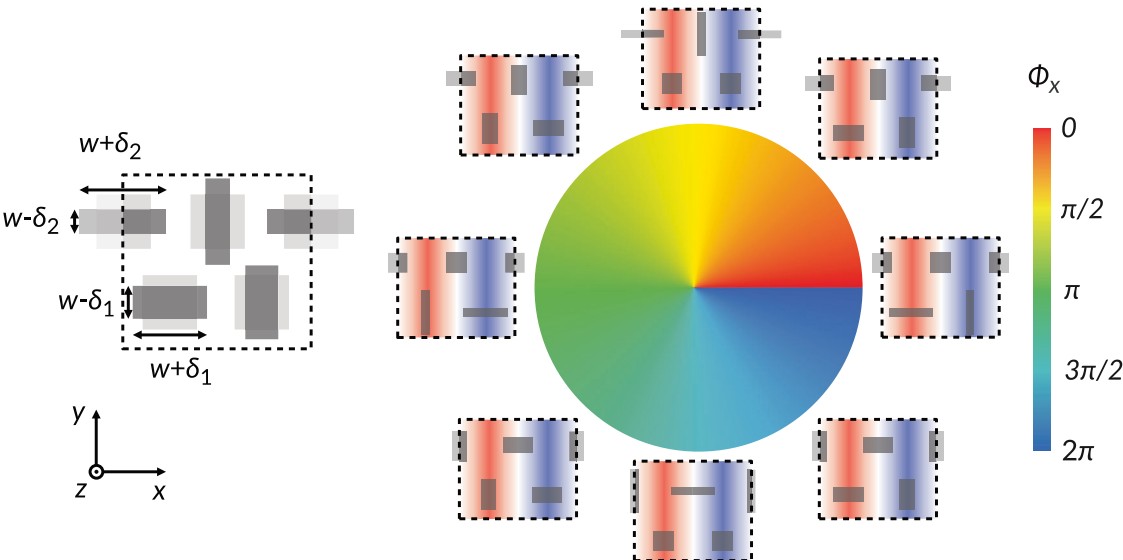

**Fig. 4 | Phase control with two symmetry-breaking perturbations.** The perturbations to the lower and upper rows can independently control the in-phase and quadrature components of the aperture, resulting in a geometric phase for $x$-polarized radiation ($\Phi_x$).

degradations typically associated with the sharp geometric discontinuities at phase wraps[33].

We note that the introduction of this final degree of freedom breaks the translational symmetry between the top and the bottom rows of the meta-units (i.e., they have shifted copies of one another), and such broken symmetry opens a bandgap at the $\Gamma$ point of the folded FBZ. However, this gap is typically very small, with negligible influence over the performance of a finite-size device.

**Full control of leaky wave radiation**

Based on these principles, we may exert full control over the four DoFs for the radiated waves from each meta-unit by imparting different perturbation strengths $\delta_{\{1,2\}}$ and rotation angles $\alpha_{\{1,2\}}$ to the lower and

upper slot pairs (see inset of Fig. 1). In turn, these DoFs allow us to tailor both the magnitudes and linear polarization states of the in-phase and quadrature components of the aperture field. More precisely, the radiated electric field of each meta-unit can be described by

$$\begin{bmatrix} E_x \\ E_y \end{bmatrix} = \delta_1 \begin{bmatrix} c_x \cos 2\alpha_1 \\ c_y \sin 2\alpha_1 \end{bmatrix} + j\delta_2 \begin{bmatrix} c_x \cos 2\alpha_2 \\ c_y \sin 2\alpha_2 \end{bmatrix} \qquad (2)$$

The weights $c_x$ and $c_y$ account for the unequal coupling strengths of the quasi-TEM mode with $x$- and $y$-polarized radiation (i.e., the Q-factor varies sinusoidally with $\alpha$). Here, we have assumed that the radiated field magnitude varies approximately linearly with respect to

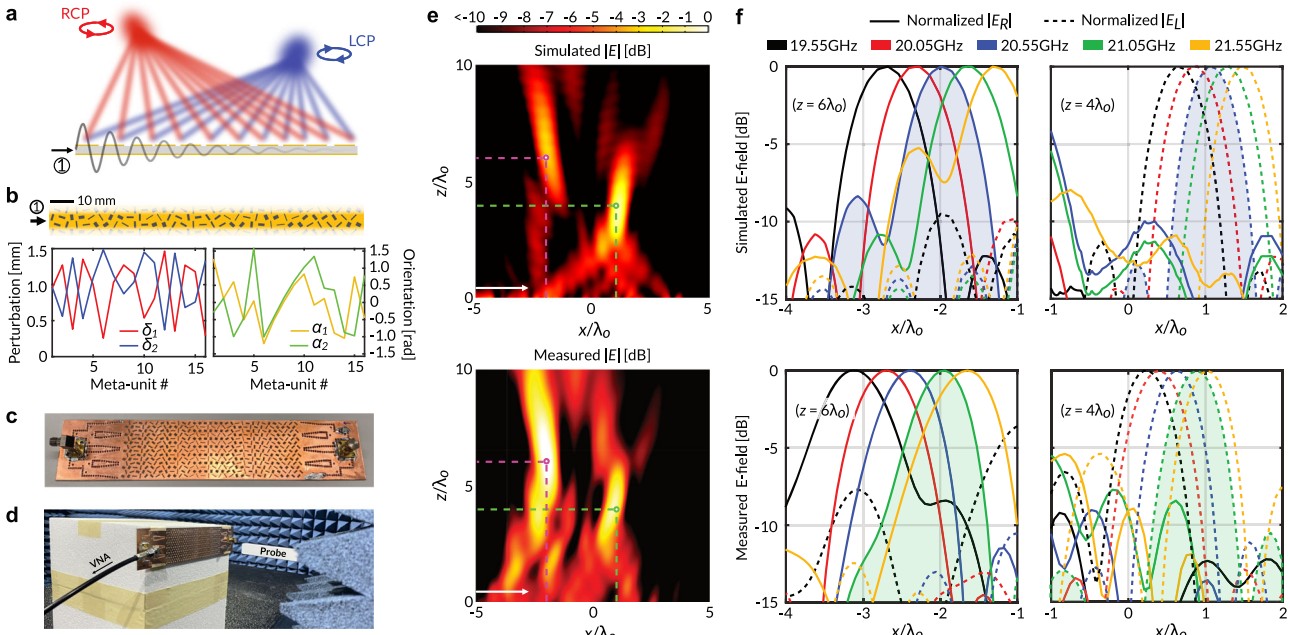

**Fig. 5 | Single-input multi-output focusing LWM. a** When excited by an input wave from port 1, the realized LWM focuses its output power to two distinct spots in the Fresnel zone, one right-handed circularly polarized, the other left-handed. **b** Analytically derived geometrical parameters for the meta-units. **c** Fabricated LWM sample. **d** Near-field scanning experimental setup. **e** Simulated (20.55 GHz) and measured (21.05 GHz) electric field intensity produced by the LWM. The pink and the green circles mark $\vec{r}_R^\star$ and $\vec{r}_L^\star$ respectively. **f** Polarization-resolved simulation and measurement results in the $z = 6\lambda_o$ and $z = 4\lambda_o$ planes confirm the correct CP state for the two focal points. As the input frequency is swept from 19.55 GHz to 21.55 GHz, the focal spots can be scanned laterally in those planes. The shaded curves correspond to the frequencies used to generate panel (**e**).

$\delta$, which is valid within the small perturbation regime and is consistent with Fig. 2g.

Equating (2) with the most general form of an arbitrarily stipulated aperture field

$$\begin{bmatrix} E_x \\ E_y \end{bmatrix} = \begin{bmatrix} A_x e^{j\Phi_x} \\ A_y e^{j\Phi_y} \end{bmatrix} = \begin{bmatrix} E_x' \\ E_y' \end{bmatrix} + j \begin{bmatrix} E_x'' \\ E_y'' \end{bmatrix}, \tag{3}$$

we obtain deterministic design equations for all geometrical parameters of the meta-unit:

$$\alpha_1 = \frac{1}{2}\tan^{-1}\left[\frac{c_x E_y'}{c_y E_x'}\right], \alpha_2 = \frac{1}{2}\tan^{-1}\left[\frac{c_x E_y''}{c_y E_x''}\right], \tag{4a}$$

$$\delta_1 = \delta \cos t_\delta, \delta_2 = \delta \sin t_\delta, \tag{4b}$$

$$t_\delta = \tan^{-1}\left[\sqrt{\left|\frac{E_x''}{c_x}\right|^2 + \left|\frac{E_y''}{c_y}\right|^2} \Big/ \sqrt{\left|\frac{E_x'}{c_x}\right|^2 + \left|\frac{E_y'}{c_y}\right|^2}\right]. \tag{4c}$$

Here, $\delta$ is the maximum perturbation strength, which can be chosen in compliance with design considerations such as overall footprint and fabrication tolerance. Note that the only system-specific parameters in (4) are the two unknown coupling constants $c_x$ and $c_y$, which can be retrieved through full-wave numerical simulations of physical models. Hence, we can use the same set of equations to design LWMs with much more sophisticated slot topologies, e.g., meandered slots, or even one with a totally different physical implementation, e.g., a grounded dielectric slab with metallic patch claddings or additively manufactured metallic pillar arrays[34]. This is in sharp contrast with conventional metasurface design methods, which heavily rely on phenomenological descriptions, such as equivalent circuits or

homogenization theory, which cannot be easily transferred between different technological platforms.

In the following examples, we adopt these rational design principles to synthesize and experimentally validate several LWM antennas with a wide range of functionalities. For simplicity, we restrict our attention to one-dimensional designs which are periodic along the $y$-direction.

### Single-input/multiple-output focusing
As one remarkable demonstration, we design a SIMO LWM antenna that generates two spatially separated near-field focused spots with orthogonal circular polarizations from a single input wave (Fig. 5a). We stipulate an aperture electric field distribution consisting of two superimposed converging cylindrical wavefronts centered at $\vec{r}_{\{R,L\}}^\star$, described by

$$\begin{bmatrix} E_x \\ E_y \end{bmatrix} = \frac{1}{\sqrt{2}}\begin{bmatrix} E_R + E_L \\ jE_R - jE_L \end{bmatrix}, \tag{5a}$$

$$E_{\{R,L\}} = H_0^{(1)}\left(k_o|\vec{r} - \vec{r}_{\{R,L\}}^\star|\right), \tag{5b}$$

where $E_{\{R,L\}}$ are the complex amplitudes of the two circularly polarized aperture field components, $H_0^{(1)}$ is the 0th-order Hankel function of the first kind, and $k_o$ is the free-space wavenumber.

We select a design frequency of 20.55 GHz in the mm-wave range and focal points $\vec{r}_R^\star = (-2\lambda_o, 6\lambda_o)$, $\vec{r}_L^\star = (\lambda_o, 4\lambda_o)$, where $\lambda_o$ is the free-space wavelength. The unperturbed square slots have widths $w = 1.85$ mm, while the maximum perturbation strength $\delta$ is set to 1.55 mm. The meta-unit dimensions are $2L_x = 8.25$ mm and $2L_y = 6.25$ mm. The $y$-periodicity is made smaller than the $x$-periodicity in order to shift all transversely propagating modes up in frequency, thereby suppressing them in the band of interest[28]. The antenna aperture consists of 16 distinct meta-units, making it approximately $9\lambda_o$ long. The substrate is

assumed to have a dielectric constant of $\epsilon_r = 3$ and thickness of $h = 1.52$mm. Materials with higher dielectric constants can be used to miniaturize the meta-unit dimensions with respect to the free-space wavelength, which serves to reduce the overall device footprint and enable more precise control over the aperture field distribution. For simplicity, we assume that the q-BIC can couple to $x$- and $y$-polarized radiation with equal efficiencies (i.e., $c_x = c_y$). Then, inserting (5) into the synthesis equations (4), we directly obtain the required meta-unit geometries of our LWM, which have been summarized in Fig. 5b.

To excite the q-BIC mode, conventional compact TEM-wave launchers can be employed[35–37]. However, they typically suffer from severely restrictive bandwidths or implementation complexity. Considering both factors, we choose to feed our LWM with a four-way substrate-integrated waveguide (SIW) power divider that tapers into a horn array (see Supplementary Fig. S1). A photograph of the fabricated prototype is shown in Fig. 5c. Figure 5d depicts our experimental measurement setup (see Methods for details). To verify the functionality of our design, we first performed preliminary full-wave numerical simulations with a simplified model of the LWM, which is infinitely periodic along the lateral direction (see Methods). The calculated normalized Fresnel-zone electric field intensity is plotted at the top of Fig. 5e. There are two clearly observable focal spots centered at $\vec{r}_R^\star$ and $\vec{r}_L^\star$, marked by the pink and green circles, respectively. Similar results can be observed in the experiment (bottom plot). Due to nonidealities such as lateral truncation, an imperfection in the input wavefront, fabrication tolerances (see Supplementary Fig. S3 for additional discussion), and deviation of the dielectric substrate properties from their nominal values, the optimal operating frequency was upshifted from 20.55 GHz to 21.05 GHz. Hence, we have plotted the measured field intensity at the latter frequency. Despite the slight discrepancy between simulation and measurements, we can observe two high-field-intensity regions near the designated focal spots. The measured −10 dB input reflection bandwidth is approximately 2.3 GHz (see Supplementary Fig. S2), which may be further improved by optimizing the transition between the launcher and the LWM.

To resolve the polarization state of the LWM output, we evaluate the LCP and RCP components at the two designated focal planes, according to[38]

$$\begin{bmatrix} E_R \\ E_L \end{bmatrix} = \frac{1}{\sqrt{2}} \begin{bmatrix} E_x - jE_y \\ E_x + jE_y \end{bmatrix}, \tag{6}$$

In Fig. 5f, we plot the normalized simulated (top) and measured (bottom) intensities of RCP (solid lines) and LCP (dotted lines) fields for various input frequencies. As expected, the focal spot can be continuously scanned in the $x$-direction simply by tuning the operating frequency, owing to the radiation mechanism of the leaky wave. Importantly, except for a single measurement (19.55 GHz, $z = 6\lambda_o$), we observe outputs with very high polarization purity, exhibiting at least 15 dB cross-polarization discrimination at the focal spots. This corroborates our ability to independently control the amplitude and phase profiles of the two orthogonally polarized field components. The lateral displacement between the simulated and measured focal spots is again attributed to the nonidealities in the fabricated sample.

We note that the experimentally measured size of our focusing LWM spot is larger than an ideal diffraction-limited spot. This is because the tapered aperture profile, which we have stipulated through (5b), serves to broaden the focal spot. As shown by Fig. S4 in the Supplementary Material, considering this factor, the focusing power of our LWM is very close to the theoretical predictions.

## Multi-input multi-output focusing

To further exploit the full aperture customizability offered by our nonlocal LWM, we implement a device that focuses its output to different arbitrary near-field positions when excited from different ports. This capability enables MIMO near-field communication via spatial-division multiple access with scannable focal spots - a desirable feature for high-throughput smart radio environments powered by reconfigurable intelligent surfaces[39].

We again stipulate an aperture field distribution consisting of two superimposed orthogonal circularly polarized cylindrical waves. However, we now assign a negative focal length for the LCP component (i.e., $y_L^\star < 0$), and encode a diverging phase front described by

$$E_L = H_0^{(2)}(k_o|\vec{r} - \vec{r}_L^\star|), \tag{7}$$

where $H_0^{(2)}$ is the Hankel function of the second kind. The RCP component of the aperture field remains the same as that given in (5). Effectively, this means that when the leaky q-BIC is excited from port 1 of the LWM, an RCP wave focused at $\vec{r}_R^\star$ and a diverging LCP wave emanating from a virtual source at $\vec{r}_L^\star$ will be generated. This results in a region of high field intensity in a right-handed CP state and weak background radiation with left-handed CP (Fig. 6a).

On the other hand, when the LWM is excited from the opposite port, a time-reversed copy of the q-BIC is excited, leading to two key consequences. First, the radiated LCP and RCP components are exchanged as they are linked to each other by time reversal. In other words, the aperture profile previously encoded for LCP is now responsible for the RCP radiation, and vice versa. Second, the patterned phase profiles are conjugated due to time-reversal symmetry, transforming a converging (diverging) wavefront into a diverging (converging) one. The combined result is that the radiation of the LWM again consists of a focused RCP spot and a weak LCP background (Fig. 6b). Notably, the focal points produced by port 1 and port 2 excitations can be independently stipulated, as they are controlled by two completely uncorrelated aperture field profiles. This allows the LWM to be used as a MIMO transceiver which communicates with sources at different locations using different ports.

In contrast with conventional techniques to achieve this effect, such as the physical partitioning of a single shared aperture[40], our proposed concept trades efficiency for more effective aperture illumination. Since each output spot fully leverages the entire physically available aperture area, we obtain significantly enhanced spatial resolution while performing near-field focusing, albeit with reduced polarization purity and power efficiency due to the cross-polarized background radiation.

Interestingly, the physical mechanism by which we achieve MIMO operation is reminiscent of the well-known phenomenon of spin-momentum-locking, which has been reported in metasurfaces with broken rotational symmetries supporting circularly polarized chiral surface waves with out-of-plane spin[41]. In our platform, the momentum (direction of propagation of the guided mode) is not necessarily locked to a circularly polarized state (spin). Instead, the customized symmetry-breaking perturbations allow us to couple the q-BIC with any arbitrary polarization state on the Poincaré sphere, such that when its momentum is reversed, the generated output field can still provide useful functionality.

We designed and fabricated a MIMO LWM prototype with the same operating frequency and key geometrical features as the previous example. The focal spots are chosen as $\vec{r}_R^\star = (-2\lambda_o, 5\lambda_o)$ and $\vec{r}_L^\star = (2\lambda_o, -6\lambda_o)$. The analytically derived meta-unit shapes are presented in Fig. 6c. In Fig. 6d, we plot the simulated and measured near-field electric field intensities with port 1 and port 2 excitations, respectively. For each input, a distinct, focused spot around the intended location (marked by the green circle) is generated, confirming the MIMO functionality of our LWM. Since the optimal frequency was downshifted by nonidealities in the prototype, we have plotted the measured fields at 20.05 GHz and the simulated fields at 20.55 GHz. Nevertheless, we observe an excellent qualitative agreement between

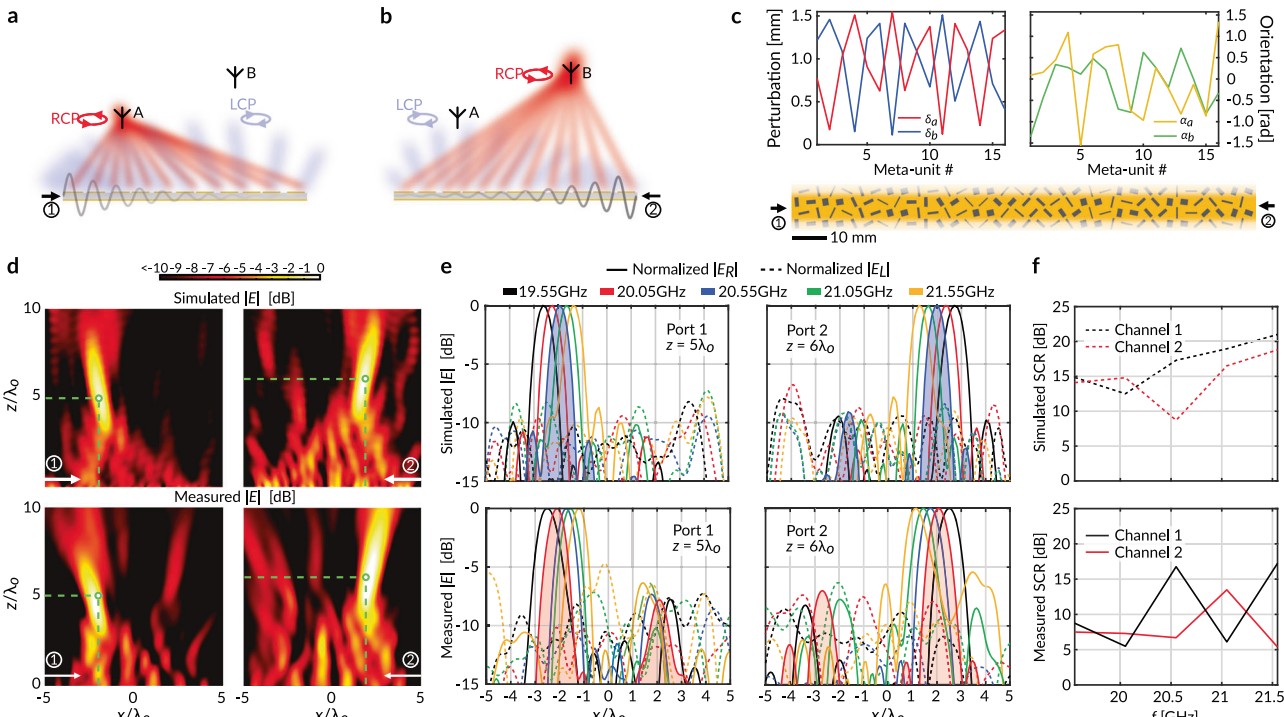

**Fig. 6 | Multi-input multi-output focusing LWM. a** When excited by an input wave from port 1, the LWM produces a focused RCP wave. **b** When excited from port 2, the same device emits another RCP wave focused on a different spot. In both cases, there is a weak LCP background. **c** Analytically derived geometrical parameters for the meta-units of the MIMO LWM. **d** Simulated and measured Fresnel-zone field electric field intensities when the two ports of the LWM are excited individually. The white arrow denotes the excitation port, while the green circle denotes the theoretically predicted output focal spot. **e** Simulated and measured electric field intensities at the two designated focal planes, demonstrating frequency-scanned focal spots with high polarization purity. The solid and dashed lines denote the RCP and LCP components of the electric field, respectively. **f** Simulated and measured signal-to-crosstalk ratio, showing good isolation between the two independent channels in a half-duplex.

experimental and simulated results, which further confirms the practical viability of our proposed LWM and feeding architecture.

Next, we examine the polarization state of the outputs in the two prescribed focal planes. In Fig. 6e, we plot the RCP (solid) and LCP (dashed) components of the simulated and measured electric field for various input frequencies at the planes $z = 5\lambda_o$ and $z = 6\lambda_o$. As predicted by theory, while one of the two ports is excited, we observe a region of high RCP field strength in its corresponding focal plane. The position of the spot can be steered by tuning the frequency. There is a weak LCP background which is at most −5 dB below the peak RCP intensity. The polarization purity is significantly improved at the center of the focal spots where the cross-polarization level is at most −9.3 dB (−9.7 dB) below the co-polarized radiation, for port 1 (port 2) excitation, throughout the measured frequency range.

To quantify the isolation between the two channels of our MIMO LWM in half-duplex operation, we define the signal-to-crosstalk ratio (SCR) for channel $i$ as

$$\text{SCR}_i = 10 \log_{10}\left(\frac{P_{i,A}}{P_{i,B}}\right), \qquad (8)$$

where $P_{i,j}$ is the power received (sent) by port $i$ from (to) an RCP transmitter (receiver) at location $j$. As indicated in Fig. 6a and Fig. 6b, $j \in [A,B]$ denotes one of the two prescribed focal spots of our LWM. The measured and simulated SCR of our LWM for various input frequencies are shown in Fig. 6f. We observe satisfactory inter-channel isolation in the measurements (solid lines), with at least 5 dB SCR throughout the entire measured scan range. The simulated SCR (dashed lines) is much better, suggesting that the inter-channel interference may be alleviated by considering the effects of the realistic SIW wave launchers, which were not included in the numerical

model. Specifically, due to imperfect wave impedance matching, the launcher on the terminating side of the LWM can generate significant unwanted reflections, which act as input from the opposite port. This effect is especially evident in the bottom left plot of Fig. 6d, which shows that port 1 excitation resulted in a weak signature at the focal spot assigned to port 2. Besides improving launcher-to-metasurface transitions, we may increase the aperture size or the leakage rate, ensuring that the q-BIC radiates most of its power before reaching the end. This solution may also unlock full duplex operation by isolating the two ports. Finally, the non-ideal phase front of the excited q-BIC, and the edge effects introduced by the lateral truncation of the sample, both serve to degrade the performance of our practical LWM prototype.

## Dual-polarized multi-beam LWM antenna
In the final design example, we demonstrate an LWM antenna capable of generating multiple independent, arbitrarily directed high-gain beams with custom polarizations. It can be useful for spot beam coverage in satellite communications, especially when paired with reconfigurable elements to facilitate dynamic beam-steering and beam-shaping[21,42].

A schematic of the proposed antenna layout is shown in Fig. 7a. We consider an illustrative design producing two RCP beams directed at $\theta_{R,1}$ and $\theta_{R,2}$, as well as a single LCP beam directed at $\theta_{L,1}$. For simplicity, we attribute each beam to a uniform aperture with an appropriate linear phase gradient. Furthermore, we assign different weights ($A_{R,1}$, $A_{R,2}$, $A_{L,1}$) to the three summands of the complete aperture to scale their relative gain in the far field. We assume the parameters $f = 20.55\,\text{GHz}$, $\theta_{R,1} = -\theta_L = 10°$, $\theta_{R,2} = 30°$, $A_{R,1} = A_L = 1$, $A_{R,2} = 1.14$, for which the synthesis equations yield an antenna design described by Fig. 7b. The experimentally measured radiation patterns match simulation results (see Methods) almost exactly in terms

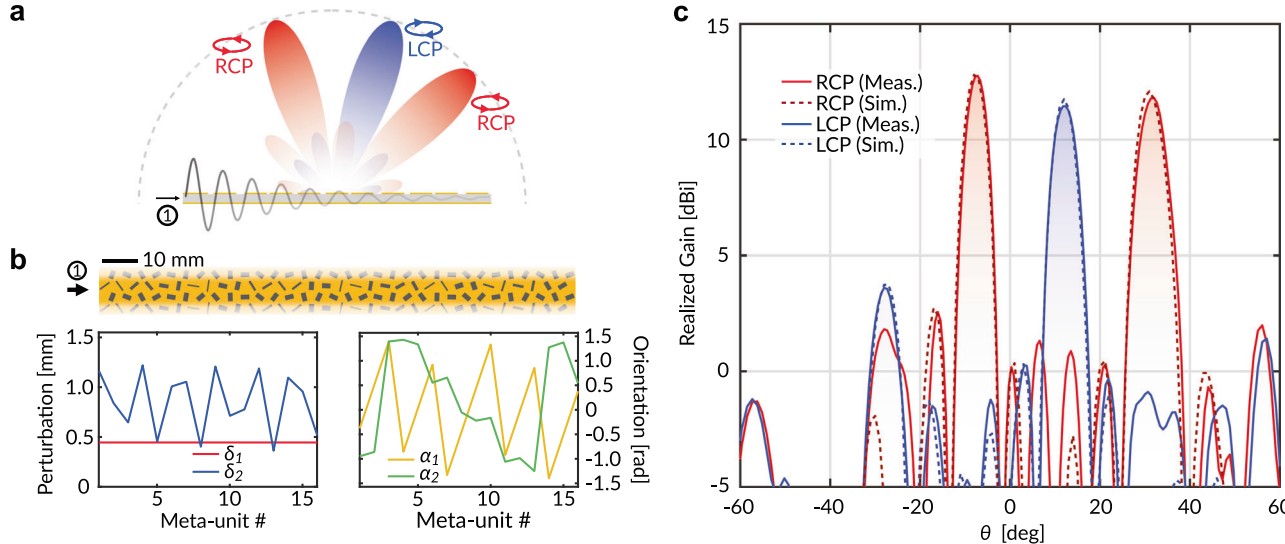

**Fig. 7 | Dual-polarized multi-beam LWM antenna. a** When excited at port 1, the LWM radiates two RCP beams, and one LCP beams with arbitrarily specified directions and relative gain. **b** Analytically derived geometrical parameters for the meta-units. **c** Simulated and measured (realized) gain of the LWM antenna.

of the realized gains and the directions of the main beams, at a slightly detuned optimal frequency of 20.8 GHz. In both results, we observe two distinctive RCP beams with −10 dB side lobe levels (SLL), as well as a single LCP beam with a −8dB back lobe at around −30°, but otherwise low SLL. The back lobe is caused by the counter-propagating wave due to reflections at the transition between the LWM and the SIW power divider. Hence, to suppress it, one can optimize the impedance matching between these two components. It is possible to further enhance the directivity of the main beams by numerically optimizing its aperture field profile, with the objective of suppressing all spurious lobes. In both the simulation and the experiment, a small amount of beam squinting from the prescribed scan angles is observed due to frequency dispersion.

In this work, we limit ourselves to the small perturbation regime for simplicity. Hence the antenna designs may suffer from reduced efficiency. For instance, the numerically simulated radiation efficiency (including dielectric and conduction loss) of the multi-beam LWM is 44%. This metric can be significantly improved by: (1) increasing the physical size of the antenna aperture; and (2) utilizing more sophisticated meta-unit designs with lower radiative Q-factors, such as a set of four meandered slots[43]. However, for each given meta-atom design, there exists a maximum bound on the leakage rate since the size of the slots and the allowable perturbations are limited by the meta-unit footprint. Hence, in order to obtain a certain level of radiation efficiency, the physical size of the LWM aperture must exceed a corresponding lower limit (i.e., it must contain sufficiently many meta-atoms). Additionally, it is important to keep in mind the inherent trade-off between radiation efficiency and aperture efficiency since large leakage rates will lead to lower q-BIC power density near the end of the LWM. This issue can be corrected using an analytically derived aperture amplitude envelope to counterbalance the tapered excitation intensity[1,44]. Finally, we emphasize that our perturbative design framework assumes a primarily dipolar-like element factor for all meta-units, which is most accurate near the broadside. At very large oblique angles (i.e., toward the end-fire directions), the presented design formulae lose accuracy in their ability to capture the nature of the radiation. To address this issue, future work can rigorously model higher-order multipole terms in the element factors[45].

## Discussion

In this work, we introduced a class of LWM antennas empowered by quasi-bound states in the continuum and associated nonlocal phenomena. By judiciously breaking and tailoring relevant symmetries in the constituent meta-units, we can independently engineer the amplitude, phase, and polarization state of the aperture fields in a pointwise manner, thereby enabling full customization of the radiated fields (see Table S1 in Supplementary Information for comparison with prior arts). The proposed framework admits an almost fully rational design scheme that eschews the cumbersome numerical optimization routines or look-up tables required by conventional approaches. Using this concept, we realized several metasurface prototypes with different functionalities, ranging from near-field wavefront shaping to far-field beam forming, all of which were validated through numerical simulations and experimental measurements. In contrast to conventional LWA approaches, which judiciously choose the position and periodicity of scattering elements in order to control the output phase profile (i.e., an aperiodic tiling), our approach uses a fixed periodicity with a spatially varying geometric phase. Future work may explore the combination of both approaches in order to create custom aperture fields with dispersion control.

We envision a straightforward application of the presented design principles to facilitate arbitrary two-dimensional aperture engineering. The main challenge is the implementation of compact and wide-band wave launchers, which can uniformly distribute the input power of a localized feed into a wide aperture. Besides conventional solutions, such as substrate-integrated reflectors, one potential alternative is the utilization of multiple independent inputs[21], which also has the added benefit of enabling fixed-frequency beam scanning in the yz-plane with the incorporation of phase shifters. It is also worth noting that all the designs discussed in this paper utilize a single guided q-BIC mode. We can augment an LWM with a suite of additional functionalities by leveraging multiple orthogonal q-BIC modes[46]. For instance, it is possible to at least double the number of independent channels in our proposed MIMO LWM with just one added mode (along with its time-reversed counterpart). Overall, our results demonstrate the power of q-BICs in a PCB-compatible radiofrequency platform and encourage further investigation of controlled symmetry as an attractive ingredient for engineering advanced microwave and mm-wave wireless devices. The generality and elegance of the symmetry-based principle introduced here also invite exploration into the terahertz regime, in which physics-informed design principles are particularly welcome due to strict technological limitations.

## Methods

### Numerical simulations

The band diagrams in Fig. 2 were obtained using the eigenfrequency solver in COMSOL Multiphysics (Electromagnetic Waves, Frequency Domain interface). A single meta-unit, consisting of a lossless dielectric substrate sandwiched between two perfect electric conductors (PEC) surfaces, the top one being perforated and both with infinitesimal thickness, is placed between two sets of Floquet periodic boundaries. The bottom side of the simulation domain is terminated by the ground plane, while the top side is implemented as a scattering boundary. Such a setup will occasionally produce spurious eigenmodes that are not supported by the metasurface but rather by the simulation domain itself. These modes can be easily identified by their non-physical power densities (e.g., growing intensities away from the surface) and have been rejected from our results.

To perform efficient preliminary confirmation of the functionality of the SIMO and MIMO-focusing LWM, we conducted simplified full-wave numerical simulations using the frequency domain solver in COMSOL. In the analysis, the LWMs were made infinitely periodic along the transverse $y$-direction using the "continuity" boundary condition. The LWM is again implemented as a lossless dielectric slab sandwiched by two PEC sheets, the top one being perforated. The two ends of the LWM were connected to short sections of unperforated PPWG. To excite the surface, the waveguide corresponding to port 1 is fed by an ideal TEM wave (launched by a uniform wave port), while the other side is terminated by a matched uniform wave port.

To accurately predict the realized gain of the multi-beam LWM antenna, we use ANSYS High Frequency Structure Simulator (HFSS) to analyze the full prototype, consisting of a laterally truncated LWM (with seven identical columns), as well as the realistic feeding networks and radio frequency (RF) connectors. In the model, the antenna is built on a lossy dielectric substrate ($h = 1.52$ mm, $\epsilon_r = 3$, $\tan\delta = 0.001$), with conductive copper claddings (thickness 0.018 mm, conductivity $5.8 \times 10^7 S/m$). The SIW horn arrays were modeled as copper via posts. The radiation efficiency of the antenna was evaluated using the radiation surface integral method.

### Device fabrication

All three designs presented in this article were fabricated in-house using LPKF prototyping systems (Protomat S104, Protolaser S4, Contac S4) on Rogers RO3003 substrates with a thickness of 1.52 mm, dielectric constant $\epsilon_r = 3$, and loss factor $\tan\delta = 0.001$.

For each sample, seven identical columns of meta-units were etched onto the substrate and repeated along the transverse direction to form a quasi-1D LWM. Ideally, a sample should contain as many columns as permitted by the device footprint, which would serve to suppress the guided wave components with non-zero transverse wavevectors, thereby minimizing the edge effects associated with the lateral truncation. However, the required SIW wave launcher would be substantially more complex due to the expanded aperture. In our simulations and experiments, we observed that seven columns provided a good compromise between device complexity and performance.

The SIW horn array launchers were fed by flange-mount SMA connectors (Amphenol P/N: 2933–6004), whose pins have been trimmed to a length of 1.52 mm to match the substrate thickness.

### Measurement

The Fresnel-zone field distributions of the SIMO and MIMO-focusing LWMs were measured using a near-field planar scanner from NSI-MI Technologies. The near-field radiation profiles of the prototypes were sampled with a K-band open-ended waveguide probe placed 5 cm away from the metasurface aperture. Then, the fields were projected to various planes parallel to the measurement plane, yielding the field intensity holograms reported in Fig. 5 and Fig. 6. The far-field radiation pattern of the multi-beam LWM antenna was obtained using the same setup. Then, its gain was evaluated by direct comparison with a K-band standard gain horn antenna (Pasternack PE-9852/2F-15, 15 dBi gain) in a separate far-field measurement setup.

During measurement, the input port of the prototype is connected to a vector network analyzer, while the opposite port is terminated by a 50Ω-matched load.

## Data availability

Authors can confirm that all relevant data are included in the paper and/or its supplementary information files, and raw data are available upon request from the corresponding author.

## Code availability

The codes used to produce these results are available upon request from the corresponding author.

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

## Acknowledgements

This work acknowledges support from the U.S. Army/ARL via the Collaborative for Hierarchical Agile and Responsive Materials (CHARM) under cooperative agreement W911NF-19-2-0119, the Air Force Office of Scientific Research MURI program with grant No. FA9550-18-1-0379, the National Science Foundation, and the Simons Foundation.

## Author contributions

A.O. developed the theoretical framework. G.X. and Y.K. designed and optimized the prototypes. G.X. performed numerical simulations and fabricated and experimentally characterized the prototypes. G.X. led the writing of the paper with contributions from A.O., Y.K., E.M., S.M., and A.A. A.A. conceived and supervised the project.

## Competing interests

The authors declare no competing interests.
