## [Peer review file · Nature Communications]

REVIEWER COMMENTS

Reviewer #1 (Remarks to the Author):

This paper reports an interesting contribution to the field of advanced antennas based on a leaky-wave metasurface approach, which allows high levels of design flexibility and functionalities to be attained. Several validation designs are presented, and one demonstrative prototype is built and tested. On the other hand, the following comments may be taken into account by the authors to further clarify their contribution:

- This reviewer misses some quantitative comparison (possibly in the form of a Table) with other state-of-the-art antennas that may exhibit related performance in terms of functionalities to better highlight the merits of this design. Even if the presented solution outperforms them, summarizing its merits in a Table would serve the reader to better catch how this work advances the prior-art.**
- For the developed prototype, the agreement between simulations and measurements is claimed as reasonable. This could be further (visually) evidenced in Fig. 5 by representing the simulated E-field distributions along with measured ones at least for some frequencies. Additionally, from an engineering viewpoint, it would be helpful to know how robust the design is to manufacturing tolerances, in the sense if a very-sophisticated manufacturing process is needed.**
- Some additional information about the design limits of this antenna approach would be appreciated.**

Reviewer #2 (Remarks to the Author):

This is an interesting paper, and I recommend its publication in Nature Communications. However, before publication, I have some recommendations for the authors.

One of the key aspects of the article is the use of multiple slots in one single unit cell. If one looks at Fig. 2a, the unit cell possesses glide symmetry with the respect to a vertical plane located in the x-axis. As a proof of the existence of glide symmetry, the authors have a dispersion diagram which is mirrored in the first Brillouin zone, i.e. a connection between first and second mode is produced at the end of the first Brillouin zone. In addition to this point, the canonical case in Fig. 2e may possess a kind of one-dimensional twist symmetry since the slots are translated and rotated. I understand that after this point, the authors have freedom for the units, and the symmetry is broken; so this is not a work about higher symmetries. However, it could be worth to mention this connection at the beginning of the paper.

In the experiments represented in Figs. 5 and 6, the resolution does not seem to be near to the diffraction limit. It could be interesting for the readers of Nat. Comm. an evaluation by the authors of the resolution that they have achieved. Also, a comment, and if possible, with simulated results (possibly in the supplementary information), showing the limitations of the technique when the leaky-wave increases in size, i.e. more resolution can be achieved.

In the Supplementary Information, there is a major incorrection that must be amended. The return loss cannot be negative. In general, negative loss means a gain, and a passive device cannot have gain. The authors must change the term "return loss" to "reflection coefficient" or simply S11. Alternatively, if the want to keep the term "return loss", they should flip up the graph for positive values of the dBs.

Reviewer #3 (Remarks to the Author):

The authors have proposed leaky-wave metasurface antennas operating at micro/millimeter-wave frequencies designed based on the principle of quasi-bound states in the continuum. Although polarization control by tilting a metallic aperture is a well-known technique in slot antenna arrays, I think the importance lies in the fact that it also allows us to decouple the phase of the local leakage from its position by engineering the geometry of a unit-cell composed of four adjacent apertures. This is distinct from the conventional leaky-wave antennas, in which the phase is fully governed by the position of the aperture.

While I personally do not see much significance to contextualize the design principle in terms of symmetry breaking or quasi-bound states in the continuum because Fig. 3 and 4 can be well explained within the conventional leaky-wave framework, I admit that the result is new, to my knowledge, and useful.

1. The word "nonlocal" is frequently used throughout the manuscript, including the title. I am not quite convinced to use this word to explain the proposed leaky-wave antenna. It is true that each aperture is mediated by a guided-wave, but the guided-wave is "weakly perturbed" by leakage as stated by the authors in page 7. Such "local" leakage from each part of the surface is not strongly mutually coupled, i.e. even if some of the apertures are disabled, the radiation pattern does not change drastically. Thus, I am afraid that the word of "nonlocal" could be even misleading.

2. Fig.4 explains the capability of arbitrary phase generation from a unit-cell. However, how to make an array of such a cell in 2D both in the x- and y-directions is not explained in detail. It should involve some additional considerations. For example, when comparing two cases in which two adjacent cells aligned in the y-direction have a phase combination of (a) 0 deg and 90deg and (b) 0deg and 270deg (as depicted in Fig.4), the distance along the y-direction between the radiative apertures becomes closer in (a) than (b). This might have an impact on the far-field pattern in the y-direction. Apart from that, a more common problem of grating lobe suppression is not discussed in detail. Unless the unit-cell is infinitely small, which is not practical, I think we should consider such additional effects.

3. I think the choice of the proposed aperture design with respect to a given far-field pattern would not be determined uniquely. In other words, the best design should be chosen out of possible candidates based on some metrics. Let me take an example of directional beamforming. In conventional slow-wave leaky-wave antennas, a periodic aperture or a grating is always used to obtain a linear phase-front. However, in the current case, the radiation phase of each unit-cell can be decoupled from its position. Therefore, we do not necessarily have a periodic aperture; even an aperiodic aperture array can generate a linear phase-front in free-space, which might show different frequency response. I would like the authors to discuss how the final pattern should be determined.

Response to the Reviewers' Comments

Reviewer #1 (Remarks to the Author):

This paper reports an interesting contribution to the field of advanced antennas based on a leaky-wave metasurface approach, which allows high levels of design flexibility and functionalities to be attained. Several validation designs are presented, and one demonstrative prototype is built and tested. On the other hand, the following comments may be taken into account by the authors to further clarify their contribution:

We thank Reviewer #1 for their time and effort in assessing our manuscript, and their constructive feedback.

- This reviewer misses some quantitative comparison (possibly in the form of a Table) with other state-of-the-art antennas that may exhibit related performance in terms of functionalities to better highlight the merits of this design. Even if the presented solution outperforms them, summarizing its merits in a Table would serve the reader to better catch how this work advances the prior-art.

We appreciate this excellent suggestion from the reviewer. We agree with the reviewer that it is important to highlight the advancements presented by the proposed technique. Hence, we have added **Table 1** to the revised version of the Supplementary Materials, which summarizes several state-of-the-art methods for designing leaky-wave metasurface antennas, while comparing their merits qualitatively. Unfortunately, it is difficult to perform a completely quantitative comparison, since the performance metrics are heavily dependent on the specific details of the antennas, such as their geometries, aperture sizes, materials, as well as their intended functionalities and operation. Furthermore, the main objective of this work is to present a new design methodology which enables new capabilities in radiation control, rather than maximizing certain performance metrics. With **Table 1**, we aim to provide the reader with a general picture of the state of the art across a few approaches, while citing a couple of representative works for each architecture.

- For the developed prototype, the agreement between simulations and measurements is claimed as reasonable. This could be further (visually) evidenced in Fig. 5 by representing the simulated E-field distributions along with measured ones at least for some frequencies. Additionally, from an engineering viewpoint, it would be helpful to know how robust the design is to manufacturing tolerances, in the sense if a very-sophisticated manufacturing process is needed.

We agree with Reviewer #1 that the addition of simulated results would strengthen our claim. Hence, they have been added to the revised manuscript (**Fig. 5 as well as Fig. 6**).

Furthermore, we have added a new figure to the Supplementary Materials (Figure S3) which discusses the robustness of the design to manufacturing tolerances. It is evident that common issues such as over- or under-etching of the conductive patterns do not significantly alter the functionality of the leaky-wave metasurface, besides slightly detuning the optimal operating frequency. This robustness stems from the radiation characteristics of the LWM being dictated by the symmetry-breaking perturbations, which will always be preserved within standard processes, as all meta-units will be over-etched or under-etched by the exact same amount.

In the experiments, we observed slightly larger frequency detuning than demonstrated by this analysis. In addition to potential deviation of the permittivity from the supplier's quoted value, this discrepancy can be attributed to the lateral truncation of the LWM sample, which is not modelled by our perturbative q-BIC framework, or the COMSOL simulations.

- Some additional information about the design limits of this antenna approach would be appreciated.

We appreciate this excellent suggestion from Reviewer #1. We have expanded our discussions on the limitations of our design approach, as well as potential solutions.

Specifically, the following limitations are highlighted:

- 1) (Page 20) There is a maximum leakage rate for each meta-unit, since its total area limits the maximum allowable slot size and perturbation. This means there exists a minimum size requirement for the LWM aperture, if the radiation efficiency must exceed a certain threshold.
- 2) (Page 20) It is not straightforward to control the radiation towards very large off-broadside angles, since our perturbative framework assumes a primarily dipolar element factor for the meta-units (yet higher order multipoles are inevitably present). To effectively engineer the radiation near end-fire directions, it is necessary to rigorously model higher-order multipole radiation from each meta-unit, or to utilize deeply subwavelength meta-units (enabled by substrates with high dielectric constants).

Reviewer #2 (Remarks to the Author):

This is an interesting paper, and I recommend its publication in Nature Communications. However, before publication, I have some recommendations for the authors.

We thank Reviewer #2 for their positive feedback as well as their insightful comments. Following their suggestions, we have made corrections as well as amendment to the manuscript to improve its quality.

One of the key aspects of the article is the use of multiple slots in one single unit cell. If one looks at Fig. 2a, the unit cell possesses glide symmetry with the respect to a vertical plane located in the x-axis. As a proof of the existence of glide symmetry, the authors have a dispersion diagram which is mirrored in the first Brillouin zone, i.e. a connection between first and second mode is produced at the end of the first Brillouin zone. In addition to this point, the canonical case in Fig. 2e may possess a kind of one-dimensional twist symmetry since the slots are translated and rotated. I understand that after this point, the authors have freedom for the units, and the symmetry is broken; so this is not a work about higher symmetries. However, it could be worth to mention this connection at the beginning of the paper.

We appreciate the reviewer's attention to this detail. We agree, the meta-units in Figure 2 have additional symmetries that are not present in the general meta-unit. We have clarified in the context of Figure 2 (Page 10) that an additional translational symmetry is present when $row1 = row2$ (which, as the reviewer points out, sometimes appears as a type of glide symmetry), but that generally this symmetry is not present. This symmetry protects the Dirac point, which is perturbed in the general meta-unit to a small degree.

Additionally, we do not believe, from the point of view of crystallography, that the two slots being rotated copies of each other in Fig. 2e is a relevant symmetry to the scattering. Changing one

rectangle into a square, for instance, would not change the space group of a periodical tiling of the unit cell, and hence does not change the selection rules (i.e., polarization state scattered due to the symmetry breaking, as discussed in Ref 25: Overvig, A. C., Malek, S. C., Carter, M. J., Shrestha, S. & Yu, N. Selection rules for quasibound states in the continuum. *Phys. Rev. B* **102**, 1–28 (2020)). In other words, it would radiate with the same phase and polarization as the meta-unit in Fig. 2e, albeit with modified radiative Q-factor.

In the experiments represented in Figs. 5 and 6, the resolution does not seem to be near to the diffraction limit. It could be interesting for the readers of Nat. Comm. an evaluation by the authors of the resolution that they have achieved. Also, a comment, and if possible, with simulated results (possibly in the supplementary information), showing the limitations of the technique when the leaky-wave increases in size, i.e. more resolution can be achieved.

Following the Reviewer’s excellent suggestion, we have studied the focusing power of the LWM and have added a short discussion on this topic (Page 14). There are several factors which may have contributed to the broadening of the spot size in the focal plane compared to the ideal diffraction limit. Most importantly, by design, the amplitude profile of our aperture fields are not uniform. Instead, following equations (5b) and (6) in the main text, we assigned cylindrical wave profiles that have tapered amplitudes away from the center. In comparison to a flat amplitude profile, this reduces the effective aperture size, and hence broadens the attainable focal spot. Furthermore, the decay of the power density of the q-BIC as it travels along the LWM will cause further deviation of realized aperture profiles from design.

To examine the realizable resolution of our LWM, we perform COMSOL simulation with a right-handed circularly polarized antenna, which focuses to a spot $(x, z) = (0, 5\lambda_0)$ at 20.55GHz. The electric field intensity in the focal plane $z = 5\lambda_0$ is plotted in Figure S4 of the revised Supplementary Materials. For comparison, we also plot the theoretically predicted field distribution produced by a continuous aperture of the same physical size ($9\lambda_0$), with the same tapered cylindrical wave profile. It is evident that our LWM can perform close to the ideal limit.

In the Supplementary Information, there is a major incorrecion that must be amended. The return loss cannot be negative. In general, negative loss means a gain, and a passive device cannot have gain. The authors must change the term “return loss” to “reflection coefficient” or simply S11. Alternatively, if the want to keep the term “return loss”, they should flip up the graph for positive values of the dBs.

We thank the reviewer for pointing out this error. The figure, its caption, as well as the relevant passages in the main text have been corrected.

Reviewer #3 (Remarks to the Author):

The authors have proposed leaky-wave metasurface antennas operating at micro/millimeter-wave frequencies designed based on the principle of quasi-bound states in the continuum. Although polarization control by tilting a metallic aperture is a well-known technique in slot antenna arrays, I think the importance lies in the fact that it also allows us to decouple the phase of the local leakage from its position by engineering the geometry of a unit-cell composed of four adjacent apertures. This is distinct from the conventional leaky-wave antennas, in which the phase is fully governed by the position of the aperture.

While I personally do not see much significance to contextualize the design principle in terms of symmetry breaking or quasi-bound states in the continuum because Fig. 3 and 4 can be well explained within the conventional leaky-wave framework, I admit that the result is new, to my knowledge, and useful.

We thank Reviewer #3 for their insightful comments and suggestions.

1. The word "nonlocal" is frequently used throughout the manuscript, including the title. I am not quite convinced to use this word to explain the proposed leaky-wave antenna. It is true that each aperture is mediated by a guided-wave, but the guided-wave is "weakly perturbed" by leakage as stated by the authors in page 7. Such "local" leakage from each part of the surface is not strongly mutually coupled, i.e. even if some of the apertures are disabled, the radiation pattern does not change drastically. Thus, I am afraid that the word of "nonlocal" could be even misleading.

We appreciate the subtlety of the reviewer's point, but we believe that the use of "nonlocal metasurface" is justified and in keeping with the recent developments in the related field of "nonlocal flat optics" (see the recent review article, DOI: [10.1038/s41566-022-01098-5](https://doi.org/10.1038/s41566-022-01098-5)). The term "nonlocal" being rapidly adopted by this community (see also the recent perspectives: DOI: [10.1002/lpor.202100633](https://doi.org/10.1002/lpor.202100633) and DOI: [10.1021/acsp Photonics.2c01534](https://doi.org/10.1021/acsp Photonics.2c01534)).

Here, when excited from free space, our metasurface response cannot be modeled with meta-units operating independently. Instead, the response depends on the fields and structures at distant locations due to the excitation of the traveling mode – a hallmark of "nonlocality" in the relevant metasurface field. This extended spatial response results in angle-selective responses following the dispersion of the leaky mode (as usual for leaky-wave antennas, but in contrast to conventional metasurfaces). On the other hand, the reviewer is correct to say that the leakage to and from the traveling mode is locally controlled (approximately dipolar), and that leveraging such locality is routine for creating custom wavefronts using metasurface approaches. Since our devices create custom wavefronts using dipolar responses ("metasurface") applied to traveling waves with angular selectivity ("nonlocal"), we believe "nonlocal metasurface" is the appropriate category for our device.

2. Fig.4 explains the capability of arbitrary phase generation from a unit-cell. However, how to make an array of such a cell in 2D both in the x- and y-directions is not explained in detail. It should involve some additional considerations. For example, when comparing two cases in which two adjacent cells aligned in the y-direction have a phase combination of (a) 0 deg and 90deg and (b) 0deg and 270deg (as depicted in Fig.4), the distance along the y-direction between the radiative apertures becomes closer in (a) than (b). This might have an impact on the far-field pattern in the y-direction. Apart from that, a more common problem of grating lobe suppression is not discussed in detail. Unless the unit-cell is infinitely small, which is not practical, I think we should consider such additional effects.

Reviewer #3 raises valid concerns regarding the generation of grating lobes. However, in our LWM designs, the periodicity along the x-direction is close to $\lambda_0/2$, where λ_0 is the free space wavelength at the design frequency. This means grating lobes will not be present inside the visible region unless we choose to form a beam near end-fire directions. In our study, we do not consider such extreme beam angles. The spurious lobes that we observed in measurements (e.g. in Fig. 7c) are attributed to backward-travelling reflections caused by imperfect impedance matching between the LWM and the SIW power divider, rather than the imperfect discretization of the metasurface aperture. Our original

manuscript incorrectly used the term “grating lobe” to describe these spurious lobes. The updated manuscript employs the term “back lobe” (Page 19), which is more commonly used by the antenna community. The back lobes can be significantly suppressed using properly optimized transitions between the LWM and the SIW power divider (as emphasized on Page 19-20).

It should also be noted that, the angular range of the LWM can be extended while avoiding the onset of grating lobes by using dielectric substrates with higher permittivity values, which shrinks the guided wavelength of the q-BIC, and thus reduces the size of the meta-unit compared to the free space wavelength. This is equivalent to decreasing the element spacing in an antenna array.

The reviewer also raised concerns about radiation patterns in the y -direction (i.e., the direction perpendicular). Again, this is a fair concern if the y -dimension (L_y) of meta-unit is large compared to the free space wavelength, or if we are concerned about radiation towards very oblique angles (in the yz -plane). In our study, L_y is always smaller than $\lambda_o / 2$. In fact, the scenario discussed by the reviewer is already present in our quasi-1D LWM designs. For example, we can have a meta-unit with $\Phi_x = 0$ followed by one with $\Phi_x = \pi$, or a unit with $\Phi_x = \pi / 2$ followed by one with $\Phi_x = 3\pi / 2$. In this case, the strongly radiating slots of these neighboring meta-units are positioned close to each other. In our extensive simulation and experimental studies, we have never encountered any issues caused by these specific combinations of meta-units.

3. I think the choice of the proposed aperture design with respect to a given far-field pattern would not be determined uniquely. In other words, the best design should be chosen out of possible candidates based on some metrics. Let me take an example of directional beamforming. In conventional slow-wave leaky-wave antennas, a periodic aperture or a grating is always used to obtain a linear phase-front. However, in the current case, the radiation phase of each unit-cell can be decoupled from its position. Therefore, we do not necessarily have a periodic aperture; even an aperiodic aperture array can generate a linear phase-front in free-space, which might show different frequency response. I would like the authors to discuss how the final pattern should be determined.

As Reviewer #3 correctly pointed out, there are two very different approaches of obtaining the same far-field radiation pattern. Following the conventional approach, one can engineer the periodicity of the leaky-wave antenna in order to couple a bound wave to a plane wave with the desired direction of propagation.

On the other hand, our proposed approach utilizes a fixed lattice periodicity (i.e., a fixed set of meta-atom positions and maximum size). Instead of relying on a position dependent phase, we locally engineer the phase profile using properly designed meta-atoms at the prescribed lattice sites. In our approach, since the periodicity is fixed by the desired frequency and the chosen dielectric substrate, we can indeed uniquely determine the required aperture phase of each meta-atom (up to a global phase constant, which has negligible effect on the performance of the LWM).

We do not claim that our proposed approach can outperform the traditional one. Rather, through this article, we wish to highlight its unprecedented versatility, which enables much more sophisticated beamforming and near-field wave-shaping capabilities than current state of the art. Using this approach, we can control with four degrees of freedom per unit cell the radiation pattern, offering significantly superior opportunities to conventional leaky-wave design approaches.

Finally, we agree with the reviewer that combining both approaches (i.e., by implementing our phase in aperiodic structures) is an intriguing possibility for future work, and that it may introduce advantages in particular regarding dispersion engineering of the output. We have added a few sentences in the outlook section (Page 21) discussing this point.

REVIEWERS' COMMENTS

Reviewer #1 (Remarks to the Author):

The authors have properly addressed my previous concerns, so that the revised version of this manuscript would be now suitable for publication in its current form. Thank you for your efforts.

Reviewer #2 (Remarks to the Author):

Thank you for addressing my questions.

The typo in the losses has been amended.

In the other questions, we have a disagreement in terms of effect of the symmetry in the operation of the periodic structure. Despite of this disagreement, I recommend the publication as it is.

Reviewer #3 (Remarks to the Author):

Thank you for addressing all my comments and suggestions. After clarifying the differences from the conventional LWA design, the manuscript has improved.

Response to Reviewers' Comments

Reviewer #1 (Remarks to the Author):

The authors have properly addressed my previous concerns, so that the revised version of this manuscript would be now suitable for publication in its current form. Thank you for your efforts.

Reviewer #2 (Remarks to the Author):

Thank you for addressing my questions.

The typo in the losses has been amended.

In the other questions, we have a disagreement in terms of effect of the symmetry in the operation of the periodic structure. Despite of this disagreement, I recommend the publication as it is.

Reviewer #3 (Remarks to the Author):

Thank you for addressing all my comments and suggestions. After clarifying the differences from the conventional LWA design, the manuscript has improved.

Authors' response:

We thank the reviewers as well as the editors for their time and effort spent on improving the quality of the manuscript. We are happy to read that all reviewers support publication of our work.